# Breast Tumor Metastasis and Its Microenvironment: It Takes Both Seed and Soil to Grow a Tumor and Target It for Treatment

**DOI:** 10.3390/cancers16050911

**Published:** 2024-02-23

**Authors:** Shirin Bonni, David N. Brindley, M. Dean Chamberlain, Nima Daneshvar-Baghbadorani, Andrew Freywald, Denise G. Hemmings, Sabine Hombach-Klonisch, Thomas Klonisch, Afshin Raouf, Carrie Simone Shemanko, Diana Topolnitska, Kaitlyn Visser, Franco J. Vizeacoumar, Edwin Wang, Spencer B. Gibson

**Affiliations:** 1Department of Biochemistry and Molecular Biology, University of Calgary, Calgary, AB T2N 4N1, Canada; sbonni@ucalgary.ca; 2The Arnie Charbonneau Cancer Institute, University of Calgary, Calgary, AB T2N 4N1, Canada; shemanko@ucalgary.ca; 3Department of Biochemistry, University of Alberta, Edmonton, AB T6G 2H7, Canada; dbrindle@ualberta.ca; 4Cancer Research Institute of Northern Alberta, University of Alberta, Edmonton, AB T6G 2E1, Canada; dghemmin@ualberta.ca (D.G.H.);; 5Division of Oncology, College of Medicine, University of Saskatchewan, Saskatoon, SK S7N 0W8, Canada; dean.chamberlain@saskcancer.ca (M.D.C.); nima.daneshvar-baghbadorani@usaskc.ca (N.D.-B.);; 6Saskatchewan Cancer Agency, University of Saskatchewan, 107 Wiggins Road, Saskatoon, SK S7N 5E5, Canada; 7Department of Pathology, Laboratory Medicine, College of Medicine, University of Saskatchewan, Saskatoon, SK S7N 5E5, Canada; 8Department of Obstetrics and Gynecology, University of Alberta, Edmonton, AB T6G 2S2, Canada; 9Department of Medical Microbiology and Immunology, University of Alberta, Edmonton, AB T6G 2E1, Canada; 10Li Ka Shing Institute of Virology, University of Alberta, Edmonton, AB T6G 2E1, Canada; 11Department of Human Anatomy and Cell Science, Faculty of Health Sciences, College of Medicine, University of Manitoba, Winnipeg, MB R3T 2N2, Canada; sabine.hombach-klonisch@umanitoba.ca (S.H.-K.); thomas.klonisch@umanitoba.ca (T.K.); 12Department of Immunology, Faculty of Medicine, University of Manitoba, Winnipeg, MB R3E OT5, Canada; afshin.raouf@umanitoba.ca (A.R.);; 13Cancer Care Manitoba Research Institute, Cancer Care Manitoba, Winnipeg, MB R3E OV9, Canada; 14Department of Biological Sciences, University of Calgary, 2500 University Dr. NW, Calgary, AB T2N 1N4, Canada; 15Department of Biochemistry and Molecular Biology, Medical Genetics, and Oncology, Cumming School of Medicine, University of Calgary, Calgary, AB T2N 1N4, Canada; 16Department of Oncology, University of Alberta, Edmonton, AB T6G 2R3, Canada

**Keywords:** metastasis, breast cancer, immunotherapy, clonal heterogeneity, bone and brain metastasis

## Abstract

**Simple Summary:**

Metastasis is one of the biggest challenges in treating breast cancer. Breast tumors grow in different areas such as the brain, lungs and bone and have distinct characteristics. The interaction between the breast tumor and its metastatic microenvironment is similar to seeds planted into soil where the cancer can grow. This review will describe characteristics of breast cancer metastasis and its corresponding microenvironment. We will also discuss evolving treatment options targeting breast cancer metastasis.

**Abstract:**

Metastasis remains a major challenge in treating breast cancer. Breast tumors metastasize to organ-specific locations such as the brain, lungs, and bone, but why some organs are favored over others remains unclear. Breast tumors also show heterogeneity, plasticity, and distinct microenvironments. This contributes to treatment failure and relapse. The interaction of breast cancer cells with their metastatic microenvironment has led to the concept that primary breast cancer cells act as seeds, whereas the metastatic tissue microenvironment (TME) is the soil. Improving our understanding of this interaction could lead to better treatment strategies for metastatic breast cancer. Targeted treatments for different subtypes of breast cancers have improved overall patient survival, even with metastasis. However, these targeted treatments are based upon the biology of the primary tumor and often these patients’ relapse, after therapy, with metastatic tumors. The advent of immunotherapy allowed the immune system to target metastatic tumors. Unfortunately, immunotherapy has not been as effective in metastatic breast cancer relative to other cancers with metastases, such as melanoma. This review will describe the heterogeneic nature of breast cancer cells and their microenvironments. The distinct properties of metastatic breast cancer cells and their microenvironments that allow interactions, especially in bone and brain metastasis, will also be described. Finally, we will review immunotherapy approaches to treat metastatic breast tumors and discuss future therapeutic approaches to improve treatments for metastatic breast cancer.

## 1. Introduction

Breast cancer (BC) is among the most diagnosed cancers in women; furthermore, it is the leading cancer-related mortality in women. In 2020, over 2 million new BC cases and more than 680,000 deaths were reported [1]. Importantly, the incidence of BC is also increasing every year. The good news is that therapies for BC are improving, and more women are living longer with BC. Nevertheless, BC cells are highly heterogeneous and interact with the surrounding microenvironment to form different subtypes with variable degrees of metastasis [2]. Consequently, this leads to distinct clinical outcomes and responsiveness to therapy.

Metastasis is characterized by the migration of primary cancer cells to distant sites in the body that then form secondary tumors. Metastatic breast tumors are notoriously hard to treat and eventually lead to treatment failure and death in many BC patients. Metastasis represents a hallmark of cancer where primary cancer cells must invade into surrounding tissue to migrate into the vascular and lymphatic vessels, survive in circulation, evade the immune system, enter into pre-metastatic sites and survive to proceed to proliferate into secondary tumors [3]. In BC, this process is governed by the heterogeneous nature of the primary tumor and changes in the microenvironment of both the primary tumor and the site of metastasis [2]. In addition, it has become evident that these pre-metastatic sites have changed to become receptive to the BC cells that have escaped the primary tumor [2]. This could be due to changes in stromal cells, immune cells or vascular permeability at the site of metastasis. This represents a concept of seed versus soil where the seed is the metastatic BC cells, and the soil is the tissue into which metastatic BC cells enter and grow [4].

In this review article, we will address the factors in BC cells (seed) leading to metastasis, define the role of the metastatic tissue microenvironment (soil) and summarize treatment strategies to target both seed and soil in metastatic BC.

## 2. Seed: Tumor Clonal Heterogeneity

### 2.1. Breast Cancer Subtypes

Traditionally, breast tumors have been classified into three clinical subtypes. The estrogen receptor (ER)+/progesterone receptor (PR) subtype represents ~70% of all the BCs. The human epithelial growth factor receptor 2 (HER2)+ subtype represents 15% of BCs, where HER2 amplification and a higher expression of HER2 are detected. Finally, triple-negative BC (TNBC) represents 15% of BCs, which lacks expression of ER, PR, and HER2. TNBC is the most heterogeneous BC with aggressive clinical outcomes. It tends to occur in younger women, recurs frequently and leads to metastasis, particularly to the lung and the brain [5].

With the development of “omic” technology, genomic and transcriptomic profiling of breast tumors have been conducted in the past two decades. These activities have provided novel insights into BC biology, profoundly influenced our understanding of BC heterogeneity and impacted patient stratification. Based on genome-wide mRNA data, BCs have been classified into five molecular subtypes [6]: luminal A, luminal B, HER2−enriched, basal-like, and claudin-low. Molecular subtypes can capture more accurately the biological, prognostic, and clinical features of tumors than traditionally used subtypes (i.e., ER+, HER2+ and TNBC). Women with luminal A tumors have more favorable relapse-free survival and overall survival among those with other breast tumors, while women with luminal B tumors have the second-best favorable relapse-free survival and overall survival. Both luminal A and B tumors are a part of the ER+ subtype, differing in the expression of PR and HER2. Basal-like tumors, representing 15% of tumors, have a higher chromosomal instability and are strongly associated with germline BRCA1 mutations. Claudin-low tumors have more mesenchymal features and poor sensitivity to chemotherapy [7]. Both basal-like and claudin-low tumors are part of the TNBC tumor subtype. We conducted genomic analysis of these subtypes based on TCGA data and found that most of the HER2+ and basal-like tumors bear somatic mutations of TP53 (70%, 90%, respectively), while most of the luminal tumors bear PIK3CA mutations (45% and 30% for luminal A and B tumors, respectively) [8].

Defining molecular subtypes supports the identification of personalized treatment for women with BC. For example, Herceptin is often used to target tumors with an HER2+ subtype. As well, most luminal A and B tumors do not need to be treated with chemotherapy, and surgical resection of the tumor is sufficient. Artificial intelligence (AI) technology has been used in evaluating luminal A and B tumors (ER+ BC) to discover gene expression signatures of those that do not need to be treated with chemotherapy. We developed a new AI algorithm and identified several gene signatures (markers) to identify which ER+ tumors do not need to be treated with chemotherapy with highly predicting accuracies (87–96%) [9]. Most importantly, different from other BC signatures, which often failed to be predictive in other independent BC cohorts, the gene signatures we identified were highly robust: they were predictive in all the public independent BC cohorts (i.e., eight independent cohorts containing more than 1000 samples) at that time [9]. For the luminal A and B tumors which need to be treated, we conducted network modeling of luminal and basal-like tumors, respectively, based on the proposed cancer hallmark network framework [10] to correctly match drugs for luminal and basal-like tumors, respectively [11]. However, the treatment of claudin-low tumors is still very challenging.

### 2.2. Clonal Evolution, Intratumor Heterogeneity and Metastasis

The different BC molecular subtypes represent a highly heterogeneous disease. In fact, heterogeneity is also found within tumors or intratumor heterogeneity, which is a key driver of metastasis. The transformation from a normal cell into a cancer cell is a gradual evolutionary process in which genomic alterations accumulate in a stepwise manner. Genome sequencing of breast tumors suggests that mutational processes evolve across the lifespan of a tumor. As the cells accumulate thousands of mutations, the developing cancer cell (i.e., the most recent common ancestor) starts to diverge into subclones of genetically related cells. We called the most recent common ancestor a ‘cancer-founding clone’. All the mutations and genomic alterations leading to the emergence of the cancer-founding clone will be carried in every cancer cell in the tumor. New genomic alterations in the founding clone will generate subclones of cancer cells. We have summarized several cancer models of subclone evolution [12,13,14]. These models suggest that tumorigenesis involves the progression from early, slow-growing subclones to late, fast-growing subclones. Although subclones within a tumor are genetically related, they gain different growth and/or invasive capabilities so that they may show different responses to therapy. Tumor genome sequencing studies [15,16] suggested that many distinct subpopulations of cells or subclones co-exist within a tumor. Genome sequencing reveals the genetic record of their emergence over time and allows us to trace the divergence of a cell to form different subclones. By the time cancer is diagnosed, one of these subclones forms the dominant population within the tumor.

As multiple subclones co-exist within a primary tumor, they have different relationships in terms of genetic profiles. We therefore summarized potential interactions among the subclones as follows [12,13]: (1) one subclone could support the growth of other subclones (for example, a subclone could amplify a ligand such as FGF, which could trigger FGF signaling pathways in other subclones); or a subclone could interact with the tumor microenvironment to protect itself and other subclones within the tumor from host immune responses; (2) one subclone could suppress the growth of another subclone by either secreting inhibiting factors or by using a larger portion of the available nutrients and growing to take over a large volume/space within a tumor; and (3) the subclones grow independently and have no interactions with each other. Multiple subclones co-exist within a tumor, representing cancer cell heterogeneity. Heterogeneity provides one of the major reasons for the failure of drug treatment in cancer.

Some subclones could undergo new genomic alterations to acquire invasive capabilities to stimulate metastasis. Analyses of the genomic data from metastatic breast tumors have shown that subclones of the metastatic lesions are derived from subclones of the primary tumor [17,18]. Subclones of metastases or recurrences, in turn, have acquired mutations and additional variants beyond the subclones in the primary tumor [19,20]. The private ‘driver’ mutations in metastases from treated versus untreated patients indicates that these changes are associated with drug resistance, but they are not associated with driving metastasis [17,21]. These results agree with our findings that the genomic alterations of founding clones largely determine if a subclone could gain invasive capabilities to generate metastasis in BC. For example, by analyzing the genome sequences of several hundred breast tumors, we identified founding clone mutations that alone were significantly associated with tumor recurrence and survival [22]. This suggests that genomic alterations, which occur before the emergence of the cancer-founding clone, are critical for the development of subclones with recurrence traits. In fact, the root of the invasive capabilities of tumor subclones for metastasis is most likely encoded in the germline genomes of the patients. For example, by analyzing 10,000 germline genomes of tumors including more than 1000 breast tumors, we revealed that germline genomic variants influence tumor somatic mutations, and they are significantly associated with tumor recurrence and survival [23,24].

## 3. Soil: The Breast Cancer Metastasis Sites and the Immune Environment

### 3.1. Metastatic Sites of Breast Cancer

Metastatic BC, which accounts for most BC recurrences, is largely associated with conventional and targeted therapy resistance [2]. Thus, metastatic BC is a major predictor of poor prognosis, morbidity and death. Cancer cells derived from primary tumors acquire many changes including becoming less adhesive and more invasive and motile. These cells can then penetrate into the surrounding normal tissue and eventually intravasate into the blood and lymphatic circulations. Some of these cancer cells, which are termed circulating tumor cells (CTCs), will survive cell death cues, exit circulation and seed into distant organs from the originating organ, i.e., mammary glands [25]. These extravasated cancer cells may remain dormant for different periods of time, until new conditions and stimuli promote their proliferation into new limited numbers of tumors or metastases. The plastic nature of particular tumor-derived cancer cells, termed tumor-initiating cells or cancer stem cell-like cells, likely enables them to acquire different cell phenotype states, which are critical for cellular events leading to their escape from the primary tumor site and the formation of new tumor masses in distant organs [26,27].

The fundamental cellular process of an epithelial–mesenchymal transition (EMT) is critical in the developing organism and contributes to homeostasis. EMT can be hijacked by cancer cells, with significance for CTC enrichment, contributing to many of the events involved in the metastatic phenotype [26,28]. Upon the successful establishment of metastases, cells undergo the reverse process of an mesenchymal–epithelial transition (MET) to promote metastatic growth. EMT has also been linked to the development of resistance to anti-neoplastic treatments [26,29]. It is becoming increasingly clear that cells transitioning between EMT and MET can give rise to the existence of cells with hybrid EMT/MET phenotypes, in which cells can co-express both epithelial and mesenchymal markers. Furthermore, this hybrid EMT/MET status has been associated with metastasis and drug resistance [30,31]. Many signaling pathways have been implicated in the induction of EMT and hybrid EMT in BC cells including the Notch, Wnt, β-catenin, hedgehog, and transforming growth factor beta (TGFβ) signaling axes, with implications for metastasis to different sites including lungs and bones [30,31,32].

Metastatic tropism depends on cancer cells from the primary tumor arriving at an organ where CTCs can seed and colonize. These cells are described as disseminated tumor cells (DTCs). Both pre-existing genetic and epigenetic factors in DTCs and the ability of these cancer cells to adapt to their microenvironment are believed to play key roles in determining organ specific metastasis [33]. Prevalent sites of BC metastasis include the lung, liver, bone, central nervous system (brain, spinal cord and leptomeninges), and lymph nodes [29]. The increased survival rates of BC patients receiving various therapies, together with the emergence of new imaging modalities, have led to an increase in the detection of metastasis to some rare sites including the oral cavity, eye, peritoneum, gastrointestinal tract and skin. Tissue-specific differences in functional cell types, immune cell populations and vascular permeability in metastatic sites affect the entry of CTCs and the development of secondary breast tumors. Attention is now being placed on how these sites of BC metastasis are different.

Extracellular vesicles play an important role in breast cancer metastasis. Breast cancer cells secrete extracellular vesicles (EVs) with cargo that alters the tumor microenvironment, allowing for metastasis to occur [33]. This could be through the activation of angiogenesis, enhancing cancer cell invasion and migration and reprogramming the extracellular matrix [34,35,36]. In addition, EVs can condition metastatic niches (soil) for breast cancer cell growth such as changing the fibroblast phenotype to a cancer-associated fibroblast (CAF) which supports breast tumor growth [37]. One cargo in EVs is miRNAs that alter gene expression in target cells. In breast cancer, miR-122 incorporates into non-tumor cells in pre-metastatic niches targeting pyruvate kinase M2 and inhibiting glycolysis, thereby increasing glucose levels for breast cancer cells [38]. Metastatic breast cancer cells secrete EVs containing miR105, increasing migration and vascular permeabilization [39]. EVs contain VEGF which increases angiogenesis and loosens tight junctions between endothelial cells allowing breast cancer cells to enter the vascular system [33]. Finally, breast cancer cell EVs induce changes in the immune system, promoting metastasis. One example is breast cancer-derived EVs activating the transcription factor NFκB in macrophages, resulting in the secretion of cytokines and the promotion of pro-inflammatory conditions [40]. This indicates that EVs are a major component of breast cancer metastasis by changing the microenvironment.

We are focusing on bone and brain metastasis because of their devastating effects on the quality of life and distinct changes in the surrounding microenvironment of BC patients. For example, breast tumors in the bone cause bone degradation, whereas brain metastasis causes cognitive impairment.

### 3.2. Metastasized Tumor-Immune Microenvironment

The immune microenvironment in each pre-metastatic site differs with respect to the balance of tissue-resident immune cells [41] and the cytokines/chemokines that these immune cells produce and respond to [42]. Unique signals from inside the organ/tissue as well as those affecting that tissue systemically continually influence these cells and factors within each organ. Although tissue-resident immune cells act as sensors and a first line of defense against localized infection and tissue damage [43,44], many of these cells also have tissue-specific non-immune functions [45,46,47] (Figure 1).

Local and systemic challenges can prepare for, rather than inhibit, circulating cancer cell entry and growth. These could include immunosuppression or inflammation induced by aging [48,49], chronic and acute infections [50,51,52,53], metabolic dysfunction [54,55], physical and mental stress [56], chemotherapeutic and/or radiation treatments [57,58,59] and immunosuppression or environmental influences including the composition of mucosal microbiota [60].

The local immune microenvironment of metastatic sites impacts both entry and metastatic cell proliferation. Metastatic cells that enter into sites can exist in a state of dormancy that is maintained by the resident immune cells, predominantly NK cells and cytotoxic T cells [61]. However, activation of the immune cells by the above-noted challenges can release this control and lead to metastatic growth. A recent article in Nature Reviews Immunology elegantly describes the immune system microenvironment in metastasis-prone tissues including the brain, bone and lungs [61].

Resident immune cells in the lungs ensure that responses to continuously inhaled foreign antigens are appropriate for the challenge. Regulatory T cells (Tregs) and the abundant tissue-resident macrophages mediate a balance between tolerance and protective immune responses by suppressing cytotoxic T cell function as needed [62,63]. When activated, lung macrophages produce IFNγ [64] and IL-1β, cytokines that are known to induce endothelial cell (EC) permeability [65]. Instead, EC permeability at these sites of metastases render the resident immune cell populations dysfunctional. EC permeability of normal functioning brains, bones and lungs differ based on the physiological requirements of these organs [66]. Increased permeability facilitates the entry of circulating cancer cells [67,68] (Figure 1).

One mechanism by which circulating cancer cells are protected from immune cell recognition and the forces of shear stress [69] is through the formation of aggregates with platelets [70,71,72,73]. Circulating cancer cells in platelet aggregates can be captured by neutrophil extracellular traps (NETs), and these NETs can then facilitate the entry of circulating cancer cells into metastatic sites [51,72,74]. Inflammation also conditions pre-metastatic sites and attracts neutrophils. Excessive NETs released by activated neutrophils increase EC permeability by degrading the glycocalyx [74,75]. These roles for NETs have been found in several metastatic sites including the lung, bone and brain [74,76]. Platelets are typically attracted to activated or damaged ECs, and so these platelet-bound cancer cell complexes will be attracted to pre-metastatic niches with this type of damage. These platelets will also facilitate extravasation by the localized release of high concentrations of the bioactive lipids, lysophosphatidic acid (LPA) and sphingosine 1-phosphate (S1P), both of which will increase EC permeability [77,78,79,80,81]. LPA also accelerates a damaging inflammatory cycle associated with tumor growth and metastasis [82,83]. S1P released from mammary epithelial cells into circulation could also behave as an attractant to circulating cancer cells similarly to the role of S1P in controlling the movement of T cells into and out of lymph nodes through S1P gradients between blood and lymph [84] (Figure 1).

The extent and impact of vascular permeability on the movement of cells into tissues is controlled by tight junctions and adherent junctions between ECs [85]. Transcellular permeability (between cells) is enhanced by numerous factors produced during inflammation and EC activation, including VEGF, TNFα and other proinflammatory cytokines [85] and bioactive lipids [77,78,82]. These factors can be produced by cancer cells, resident immune cells within organs such as macrophages, NK cells, T cells, mast cells and by non-immune resident cells such as fibroblasts, epithelial cells, or osteoblasts [86]. EC permeability is also regulated by the thickness and intactness of the vessel lumen-facing glycocalyx [87,88]. The glycocalyx layer acts as a complex barrier that limits the access of permeability factors and cells to the ECs [89]. Glycocalyx is composed of proteoglycans and glycoproteins with attached heparan sulfate and chondroitin sulfate chains and hyaluronic acid [90]. Adhesion molecules that are needed for the cellular extravasation of cells through the endothelium are below the surface of the glycocalyx and are not normally accessed by cells unless the glycocalyx is dysfunctional, which occurs under inflammatory conditions [87]. The glycocalyx of pulmonary arteries is considerably thicker than in arteries from other organs.

This may reflect a need under normal physiological conditions to carefully control the access of neutrophils to the lungs [88]. Inflammation induces the cleavage of chondroitin sulfate, heparan sulfate chains or sialic acid by specific enzymes and the cleavage of the proteins by matrix metalloproteinases (MMPs), resulting in the thinning of the glycocalyx and impaired protective functions [91]. The degradation of heparan sulfate, sialic acid and chondroitin sulfate residues in the glycocalyx increases the attachment of leukocytes and circulating cancer cells to ECs [91,92,93].

## 4. Soil: Tissue-Specific Properties at Metastatic Sites

### 4.1. Bone Metastasis of Breast Cancer (Figure 2)

BC bone metastasis is incurable. BC metastasizes to the bone occurs in approximately 70% of all cases of advanced BC [94,95]. The pattern of bone metastasis is more frequent in luminal A and luminal B subtypes based upon a large Chinese study [96]. The most common subtype associated with bone metastasis is also the ER+/HER2– subtype according to the American Surveillance, Epidemiology and End Results (SEER) database [97].

Bone metastasis can be osteoblastic, characterized by the buildup of bone, or osteolytic, characterized by the loss of bone. Bone metastasis in BC patients is primarily osteolytic, which involves the destruction of the bone by factors produced from BC cells at the site of their spread to the bone in a vicious cycle that is accelerated by prolactin (PRL). BC cells in the bone secrete factors that act on osteoclasts to induce differentiation and break down the bone or on osteoblasts, which can build up bone. The release of growth factors and Ca^2+^ from the bone matrix stimulates the proliferation of BC cells, perpetuating a vicious cycle of bone destruction (for a review, see [98], Figure 2).
Figure 2Vicious cycle of bone metastasis. Breast cancer cells within the bone microenvironment secrete a number of factors, including sonic hedgehog (SHH), which can act on the osteoblasts or pre-osteoclasts and mature osteoclasts of the bone. Lysis of the bone by the osteoclasts results in breakdown of the bone matrix and the release of growth factors and calcium, which stimulate cancer cell replication and survival. Prolactin (PRL) binds to the PRL receptor (PRLR) on breast cancer cells, leading to PRL-mediated signaling that stimulates SHH production and other unidentified factors. Created with BioRender.com.
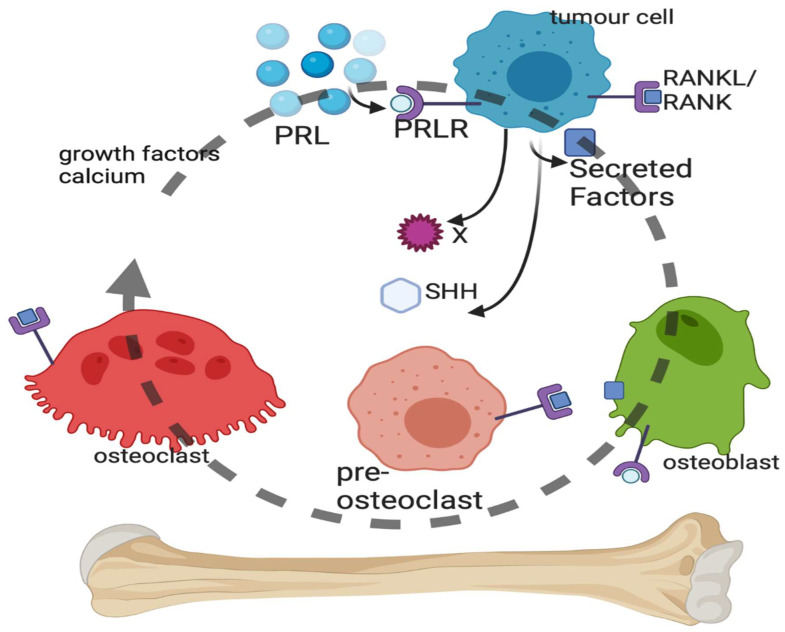


The contribution of PRL and the PRL receptor (PRLR) to BC etiology and progression is overall associated with invasive behavior [99], poor prognosis [100], increased BC cell survival [101], DNA damage resistance [102], and the induction of lytic bone metastases [103]. PRL-PRLR activation in BC cells accelerates bone metastasis [103]. Circulating cancer cells were enumerated from advanced BC patients using Veridex CellSearch and then isolated using Ficoll separation. PRLR-positive BC cells were found circulating in the blood as well as in BC bone metastases biopsies. Using a PRLR antagonist, it was shown that PRL via PRLR on BC cells accelerates osteoclast differentiation (osteoclastogenesis) and bone lysis in vitro by the production of secreted factors [103]. Osteoclasts do not express PRLRs [104], and so the PRL induction of osteoclastogenesis is BC cell-mediated.

There is a large number of cytokisnes, proteins and other molecules which can alter bone homeostasis and therefore impact a large number of osteoclast receptors. There are at least 10 cytokines known to induce multi-step osteoclast differentiation and 9 cytokines that inhibit it [105]. There are at least 28 known secreted factors from BC cells that induce osteoclast differentiation, some of which are cytokines [98]. Macrophage colony-stimulating factor (M-CSF) and its receptor encoded by the c-FMS gene and the receptor activator of nuclear factor kappa-B (RANK) ligand (RANKL) and the RANK receptor form the two main osteoclast differentiation pathways [14] with well-characterized signal transduction [106] (Figure 2).

The microenvironment of the bone is predominantly immunosuppressive with Tregs, few mature immune effector cells and large populations of myeloid progenitor cells [107]. The immune cells and their production of Type 1 interferons are mainly protecting the bone by suppressing the bone-resorbing function of osteoclasts [108]. However, while Type 1 interferons increase vascular permeability, which can promote the entry of circulating cancer cells [109], these interferons can also maintain dormancy of potentially metastatic cells [110]. The microenvironment of bone metastases in BC appears to suppress T cells, compared to the primary tumor, and this increases osteoclast formation and bone damage in osteolytic bone lesions. A syngeneic mouse 4T1 mammary cancer model was used to demonstrate that the tumor-infiltrating lymphocytes (TILs), in particular non-activated T cells, increased osteoclast formation and bone lesions. The bone metastases associated with 4T1 cells were associated with T cell-suppressing polymorphonuclear and monocytic myeloid-derived suppressor cells (MDSCs) [111]. Using primary breast tumors and their matched bone metastases, it was determined that stromal TILs (CD4+ and CD8+) were reduced in the bone metastasis compared to the primary tumor; macrophages (CD68+ and HLA-DR+) were unchanged. Programmed cell death protein 1 (PD-1) and PD ligand 1 (PD-L1) expression were strongly reduced, suggesting a less active immune microenvironment [112]. The implications of these results indicate that immunotherapy and T cell activation could be potential treatment avenues.

### 4.2. Brain Metastasis of Breast Cancer (Figure 3)

The blood–brain barrier (BBB) is a key obstacle for circulating cancer cells to enter the brain. Brain microvascular ECs are a critical component of this selective barrier [113]. Brain ECs are interconnected by tight junctions and demonstrate a very low rate of transcytosis—two properties that limit both para- and transcellular transport across the BBB [114]. Microvascular ECs are supported by a continuous basal membrane that anchors ECs and ensures apicobasal polarity. The BBB is part of the neurovascular unit [114]. Apart from microvascular ECs and the basal membrane, the neurovascular unit includes astrocytes, pericytes, smooth muscle cells, neurons, and an extracellular matrix. The complexity of reciprocal interactions between multiple components of the neurovascular unit is thoroughly reviewed elsewhere [115,116].

The initial step of brain metastasis includes CTCs arresting within the lumen of brain microvessels [117]. A small microvascular lumen size [117] and the interaction of cell adhesion molecules (CAMs) on the surface of CTCs with corresponding CAMs on brain ECs [118] are considered to contribute to this CTC arrest. In particular, BC cells expressing high levels of *MUC1*, *VCAM1*, and *VLA-4* were able to strongly adhere to brain endothelium and withstand fluid shear stress normally occurring within blood vessels [118]. In addition, breast CTCs induced the expression of E-selectin, VCAM-1, ALCAM, ICAM-1, VLA-4, and β_4_ integrin by brain ECs, demonstrating the reciprocal nature of these adherence mechanisms [119]. In addition to the cyclooxygenase COX2 and EGFR ligands previously shown to mediate lung metastasis, the 2,6-sialyltransferase ST6GALNAC5 was shown to specifically promote BC cell adhesion to brain ECs by increasing the surface expression of the ganglioside GD1α for improved trans-endothelial migration [120]. The metalloprotease ADAM8 was found to be increased in brain metastases and was shown to promote adhesion to brain ECs by releasing glycoprotein PSGL-1, a ligand of the endothelial adhesion molecule P-selectin [121]. Adhesion studies in vitro with brain ECs demonstrated that TNFα inflammatory signaling increased the expression of selected adhesion proteins in ECs (*ICAM1*, *CD112*, *CD47*, *JAM-C*) and in cancer cells (ALCAM, CD6) [122] (Figure 3).
Figure 3Breast cancer brain metastasis. (**A**) Establishing patient-derived breast cancer brain metastasis in vivo models: intracardial xenografting of breast cancer cells reflects hematogenic colonization of the brain. (**B**) Early steps of BC brain colonization involve crossing the blood–brain barrier to reach the perivascular space with contacts to pericytes, astrocyte foot processes and perivascular macrophages. (**C**) Established brain metastatic lesions may grow as demarcated spherical tumors with angiogenesis or by vascular co-option along preexisting blood vessels as observed for HER2 overexpressing (HER2+ BC) or triple-negative (TNBC) breast cancer brain metastasis, respectively.
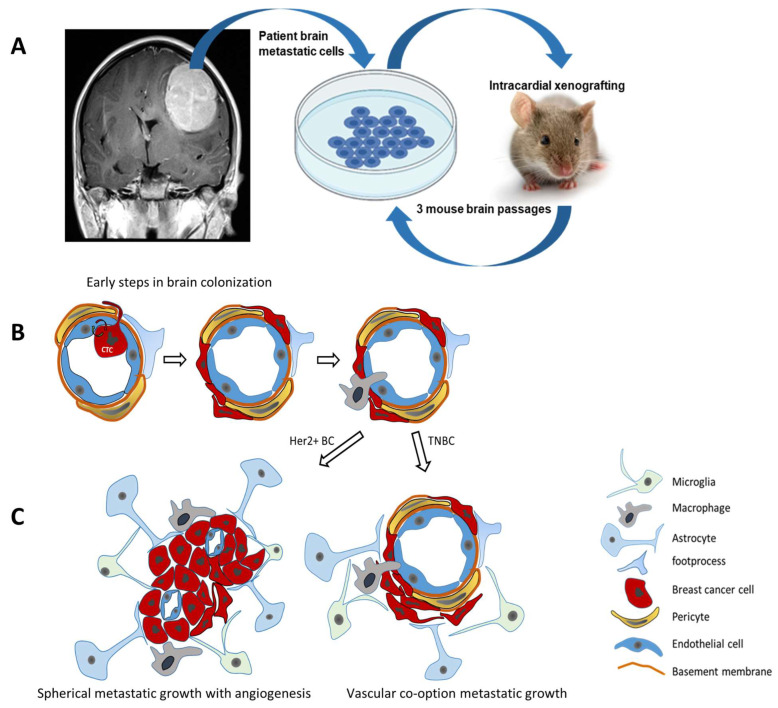


While the brain is protected by a particularly tight-junction blood–brain barrier, the brain contains or attracts the entry of several immune cells that control metastatic growth including Type 2 innate lymphoid cells (ILC2s) [123]. Microglia, the most prevalent resident immune cell in the brain, can be polarized similar to other macrophages and thus suppress [124] or allow [125] metastatic growth depending on the context of the microenvironment, regional differences and activation state. If microglia are activated, they begin to produce proinflammatory factors such as TNFα, IL-1β, IL-6, VEGF, and these can increase EC permeability and create an opportunity for the entry of circulating cancer cells [126]. Drug treatment of brain metastases is challenging as it requires access through the BBB and accumulation in the brain at effective therapeutic concentrations [127].

Direct cell-to-cell contacts are not the only way to promote the metastasis process in the brain. An activation of astrocytes was observed even prior to the extravasation of the CTCs [128]. This correlates with the complex involvement of activated astrocytes in the MMP-driven deterioration of EC tight junctions, the transmigration of CTCs, and the progression of metastatic growth in the brain [129,130,131]. The cell adhesion protein protocadherin (PCDH7) expressed by BCBM cells from the TNBC subtype interacts with connexin-43 of astrocytes, facilitates cell communication between astrocytes and BC cells and promotes BCBM growth by inducing interferon and NF-κB signaling in TNBC cells [132]. The transmigration of CTCs can potentially occur at any site of the brain microvascular bed; however, properties such as the presence of a perivascular space providing reduced resistance on the abluminal side [133,134] and the ability to accommodate the transendothelial migration of immune cells [134] favor postcapillary venules as a more susceptible site for the transmigration of CTCs. The transmigration of CTCs through the endothelial barrier also results in stripping pericytes from the abluminal surface of normal microvessels, which renders them even more leaky and penetrable by CTCs [135].

The BBB within brain metastatic tumors was shown to be leaky, and a more detailed analysis discovered regional variability in leakiness due to differences in cancer cell crosstalk with resident cells of the neurovascular unit and tumor angiogenesis [136]. The altered BBB permeability in brain tumor lesions is referred to as the brain tumor barrier (BTB) and has been associated with a high expression of the pericyte marker, desmin, and lower collagen-IV content in the EC basement membrane [137]. Contrast enhancement studies in experimental brain metastasis models of TNBC and HER2+ BC revealed a leaky BTB in more than 90% of brain metastatic lesions and a substantial heterogeneity between different metastatic lesions within brains in these patients. Interestingly, BTB permeability was enhanced for both small and large permeability markers independent of lesion size and the metastatic growth phenotype (vascular cooption or spheroid growth). Importantly, no correlation between BTB permeability and lesion size was observed for treatments with paclitaxel or doxorubicin. The measured drug concentrations in brain metastases were 10 times lower than those in peripheral metastatic tissues, emphasizing that a barrier still exists, preventing therapeutic drug levels [137]. Metastatic lesion uptake studies in patients revealed that the levels of capecitabine metabolites and lapatinib in BCBM tumors varied widely between patients and between different lesions [138]. The immunolocalization of glucose transporter 1 (GLUT1) and the BC resistance protein at the apical membrane of ECs in resected patient BCBM confirms the presence of a barrier function and suggests that the BTB of HER2+ metastatic lesions is less permeable compared to TNBC or basal-type BCBM [139].

Structural and functional interactions within the brain parenchyma are not limited to the cellular components but also involve the extracellular matrix. The composition of the brain extracellular matrix is unique and is divided into several distinct compartments [140]. While collagen is still present within the basement membrane of cerebral vasculature, the neural interstitial matrix is almost completely devoid of collagen and consists primarily of chondroitin sulphate proteoglycans, tenascins, and hyaluronan [140]. In glioblastoma, peritumoral overexpression of tenascins and hyaluronan as well as the depletion of chondroitin sulphate proteoglycans are linked to more aggressive behavior [141,142]. Despite the current lack of studies directly characterizing extracellular matrix changes in BCBM, it is well established that multiple cancers, including BC, utilize extracellular matrix remodeling to facilitate invasion and metastasis [143,144].

BC utilizes a rich repertoire of signaling pathways during tumor progression (reviewed in: [145]). This includes the canonical Wnt and Notch signaling pathways, which specifically promote the invasiveness and brain colonization of cancer cells from basal-type BCs [146,147,148]. Both pathways are essential drivers of stemness in the brain perivascular niche. The phosphorylation of the Src kinase at Y416 is increased in brain metastases compared to primary BC tumors. Experimental brain metastasis models and in vitro BBB models were used to show that Src signaling promotes brain colonization by cancer cells from TNBC and HER2+ BC through BBB disruption [149]. ErbB2 and Src kinase activity promote the downstream PI3K-AKT-mTOR pathway to enhance the growth and survival of brain metastatic HER2+ BC cells [149,150]. Not surprisingly, PI3K/AKT/mTOR activity has also been identified as the major driver of resistance to HER2−targeting therapies [151]. In BC brain lesions, but not primary tumors, the HER3 (ErbB3) receptor is gaining particular importance as emphasized by an increased expression of HER3 and HER3/HER2 downstream signaling [152]. In HER2+ BC cell lines, HER3 is the dimerization partner of HER2 and facilitates the action of brain-derived neuregulin-1 during trans-endothelial migration in vitro [153]. In summary, brain metastasis commonly occurring in TNBC and HER2+ BC differ in histopathology and molecular pathway activation which presents an urgent unmet clinical need for more efficacious brain permeable therapeutic strategies.

## 5. Treatment Opportunities for Targeted Versus Immunotherapy Approaches for Metastatic Breast Cancer

### 5.1. Seed: Targeted Treatments of Different Subtypes of Metastatic Breast Cancer Cells

Only 6% of women will present at initial BC diagnosis with metastatic disease [154]; however, up to 30% of women with BC will eventually develop metastatic disease. Adjuvant therapy is used to control the development or reoccurrence of the tumor including metastatic disease. Unfortunately, most therapies are not curative, and metastatic disease is often drug resistant [155]. This illustrates the need to develop new therapeutic strategies targeted at metastatic breast cancer.

Aromatase inhibitors have almost replaced the selective estrogen receptor modulators, such as tamoxifen, as the first-line hormone therapy for ER+ BC. Aromatase inhibitors block the production of estrogen and are used to treat postmenopausal women, as reviewed in [156,157]. Considering the rapid emergence of resistance to ER-blocking agents, selective androgen receptor modulators (SARMs) can provide us with one additional treatment option to avoid or at least postpone standard chemotherapy. Currently, the most advanced SARM member is enobosarm, which is under Phase III clinical trials and is one of the closest agents of this class to entering the market (National Library of Medicine, NCT05065411, Table 1). The combination of other targeted therapies that can be included are PARP inhibitors for patients with BRCA mutations and PI3K inhibitors for patients with activating PIK3CA mutations. Standard chemotherapy remains the final option if all these targeted therapies fail or if there is a risk of organ failure.

Similarly to ER+ BC, there are targeted treatment options for HER2+ BCs against the HER2 receptor. The first-line treatment is trastuzumab and pertuzumab, both of which are antibodies to different domains of HER2, plus a taxane chemotherapy agent, such as docetaxel. The gold standard second-line therapy is ado-trastuzumab emtansine, although the DESTINY-Breast-03 clinical trial suggests that better outcomes are seen with trastuzumab deruxtecan, but this is not currently available in all jurisdictions. Patients with brain metastases are treated with tucatinib–capecitabine–trastuzumab combined with radiotherapy. Despite the success of these monoclonal antibodies, they are only capable of targeting one epitope. Thus, this treatment will eventually be followed by alterations in the down-stream signaling pathways and the emergence of resistance. This issue is being addressed due to the fact that the clinical efficacy of DC vaccines is under investigation. This novel approach aims to induce a strong T cell-mediated immune response against cancer cells expressing HER2. In this regard, dendritic cell vaccines against HER2/3 for the treatment of TNBC or HER2+ BC with brain metastasis have been used in one of the most recent trials, which is currently in Phase IIa (National Library of Medicine, NCT04348747, 2023, Table 1).

With TNBC, there are no specific receptors to target; therefore, treatment of metastatic disease focuses on screening tumors for BRCA mutations for treatment with PARP inhibitors, or for PD-L1 expression for the use of a PD-L1 checkpoint inhibitor. Sacituzumab govitecan is an antibody that recognizes Trop-2 expressing cells, has a topoisomerase I inhibitor as a drug conjugate and is a second-line therapy, but further clinical evidence is needed. Interestingly, PIK3CA mutations are not currently considered in the treatment of TNBC. PIK3CA mutations are thought to occur in ~17% of TNBC [158], which is a similar rate to BRCA mutations within TNBC [159]. However, there are several active clinical trials looking to target PIK3CA mutations in TNBC for treatment. The combination of nab-paclitaxel with alpelisib for the treatment of TNBC cases with PIK3CA or PTEN alterations has shown promising efficacy.

Another area of active research to develop treatments for metastatic breast cancer is focused on targeting the autotaxin (ATX)–lysophosphatidate (LPA)-inflammatory cycle characterized by ATX, a secreted lysophospholipase D enzyme, which catalyzes the production of extracellular LPA from lysophosphatidylcholine. ATX is involved in increasing chronic inflammation, which in turn stimulates more ATX secretion and the subsequent increase in activation of six G protein-coupled receptors that are differentially expressed in different cells in the tumor. Overall, LPA increases cell division, survival, migration and immune suppression that promotes tumor growth, angiogenesis and metastasis [82,83]. Additionally, targeting this axis through the inhibition of ATX has been correlated with the improved efficacy of some chemotherapeutics including taxanes [160], doxorubicin [161] and tamoxifen [162]. Currently, there are no clinical trials to evaluate ATX inhibitors in the treatment of breast cancer, and so far, none of the developed ATX inhibitors have been approved as an anti-cancer agent. However, IOA-289 is now in Phase 1B trials for the treatment of metastatic pancreatic cancer (National Library of Medicine, NCT05586516, Table 1).

Metastatic BC most often forms in the bone, brain and lungs, and there are some site-specific modifications to treatment plans based on the site of metastasis. Bone and brain metastatic sites are most commonly treated with surgery and radiation therapy as first-line treatment options. Metastatic disease to the bone can cause osteoporosis; therefore, the patient is often treated with denosumab and/or a bisphosphonate drug. Other than that, the patient has the treatment as outlined above. With BCBM, the drug must cross the BBB, which not all drugs can do. Patients with progressing brain metastases most often occur in women with HER2+ BC. The only guidelines for changing treatment beyond surgery and radiation therapy are to use tucatinib, trastuzumab and capecitabine.

All the current treatments of metastatic BC still focus on treatment of the cancer cells but not the metastatic process itself. Currently, there are no successful methods to treat the metastatic process, although there have been a few attempts to do so. One example is the use of MMP inhibitors to block cancer progression. Several clinical trials were performed in the early 2000s, but these failed in Phase 3 often due to dose-limiting toxicities and the lack of efficacy [163]. Doxycycline is the only FDA-approved MMP inhibitor, which is approved as an antibiotic against infections caused by Gram-negative bacteria. The anti-cancer effects of doxycycline and other MMP inhibitors have been demonstrated by several pre-clinical studies [164], but further research is needed for the development of effective and safe MMP inhibitors for cancer therapy [165,166]. Significantly, doxycycline increases the degradation of extracellular LPA by lipid phosphate phosphatases. This effect on LPA signaling decreases the production of several inflammatory cytokines in breast tumors, the activation of NFkB and thus the inflammatory milieu of the tumor. These anti-inflammatory effects of doxycycline delayed breast tumor growth [162].

It could be argued that one of the reasons for the lack of drug development to target metastatic disease is that we have not developed strong tools to study the effects of already existing cancer therapies on this process. One discovery that may change this is the identification and quantification of CTCs in the blood [76]. There are now several different methods under active research to detect CTCs [167]. However, CellSearch is the only method that has been approved by the FDA for CTC detection [168]. CTC levels within the blood have been correlated with increased metastatic disease and poor prognosis [169,170]. Moreover, the organization (single cells versus clusters of cells) of CTCs found in the blood has been associated with their metastatic potential [171]. This suggests that studying how interventions regulate the CTC level could identify the treatments that at least disrupt the early steps of metastasis.

### 5.2. Soil: Immunotherapy Targeting the Metastatic Breast Tumor through Alteration of Its Microenvironment

The concept that the immune system can recognize and eliminate malignantly transformed cells dates back to the late 19th century when William Coley was experimenting with heat-killed bacteria preparations to induce immune cell responses and spontaneous tumor regression in cancer patients [172]. Since then, much has been learned about the role of the adaptive immune cells and, to some degree, the innate immune cells responses against cancer cells. The discovery of the first tumor-associated antigen, the melanoma-associated antigen-1 (MAGEA1), paved the way for T cell-based therapies with the idea that the patient’s own T cells can be primed to recognize specific antigens on cancer cells and eliminate them [173]. Soon after this conceptual advance, agents such as interferon-α2 and IL-2, which enhance T cell functions, were approved by the US FDA for the treatment of metastatic melanomas [174]. Based on these conceptual frameworks and other advances, different forms of immunotherapy approaches are currently being developed against solid tumors. These approaches include (A) immunomodulators, (B) adoptive cell transfer therapies and (C) cancer vaccines. We will first briefly explore each approach and then discuss ongoing clinical trials with respect to the treatment of metastatic BC tumors.

(A)Immunomodulators: Upon exposure to a tumor-specific antigen, naïve T cells differentiate into effector cytotoxic CD8+ T lymphocytes (CTLs) that recognize and eliminate cancer cells through the secretion of cytokines and degrading enzymes through cell-to-cell contact. Ultimately, these effector T cells undergo apoptosis or further differentiate into tissue-resident memory T cells [175]. To prevent the prolonged activation of T cells, the immune system has evolved to develop an inhibitory mechanism to cause T cell dysfunction and exhaustion. This mechanism was initially described in a mouse model of chronic viral infection where T cell exhaustion was found to be due to antigen overstimulation [176,177,178]. In this context, T cell dysfunction or exhaustion was caused by increased expression of “checkpoint” inhibitory receptors such as PD-1, cytotoxic T lymphocyte antigen-4 (CTLA-4) and T cell immunoglobulin domain and mucin domain protein-3 (TIM-3) on the T cells [174]. Such inhibitory receptors are activated by the expression of their cognate ligands (e.g., PD-L1) on antigen presenting cells, such as dendritic cells and macrophages. In the microenvironments of solid tumors, such as BC, T cell exhaustion is frequently observed due to the increased expression of PD-L1 on the cancer cells and increased and sustained expression of inhibitory receptors on the TILs which could then lead to CTL exhaustion [179]. Perhaps the most convincing evidence was provided from experiments showing that blocking the PD-1 interaction with its ligand PD-L1, with a monoclonal antibody, reactivated the CTLs and suppressed the growth of tumors [180,181]. Based on this and similar confirmatory data, immune checkpoint blockade using monoclonal antibodies such nivolumab and avelumab have been approved for use in the clinic to treat melanomas, Hodgkin lymphoma, and lung and other cancers. More recently, to extend the effectiveness and duration of reactivity, some patients were treated with a combination of two immune checkpoint inhibitors: one to negate the PD-1/PD-L1 interaction and another to counteract the CTLA4/CD80 or/CD86 interactions. Clinical trials are now underway to test the effectiveness of these immune checkpoint inhibitors in other solid tumors including liver cancer, non-small cell lung cancer and some BCs [182] (Table 2).

(B)Adoptive cell transfer therapies: Cellular immunotherapy or the adoptive cell therapies refer to approaches that involve isolating the patient’s own T cells and either expanding them directly or genetically modifying them to enhance their anti-cancer effector functions prior to their expansion ex vivo. These activated T cells are then reinfused back into the patient with the idea that these cells are tumor reactive and will result in tumor regression. These treatments include TIL therapy and chimeric antigen receptor (CAR) T cell therapy.

BC patients whose tumors show extensive TILs have better prognosis [183,184]. This clinical observation provided a framework to hypothesize that, at the least, some T cells in the tumor microenvironment have been primed to specifically recognize tumor-associated antigens, and therefore, their expansion ex vivo and transfer back to the patient could have therapeutic benefits [185]. The reinfusion of the patient’s expanded TILs with IL-2 showed some success in treating metastatic melanomas; however, the TILs showed a short response rate [186]. Although TIL therapy has great therapeutic potential as it gets around the autoimmune and the graft-versus-host disease immune responses, TIL therapy has shown little effect against breast and other cancer types [187]. However, recent data provided by Zacharakis et al. showed that the reinfusion of TILs from a patient with a therapy-refractory metastatic ER+HER2− tumor provided durable tumor regression. In this case, the TILs were selected based on their ability to specifically detect four different mutated proteins that were observed in the patient-specific cancer cells [188]. However, the sustained reactivity of these TILs was achieved in combination with immune checkpoint inhibitors and IL-2 to activate the TILs [188].

With the recognition that activated CTLs can specifically identify and eliminate cancer cells, cell engineering techniques were employed to generate T cells that recognize cells expressing tumor-associated antigens while sparing the normal cells. These antigens can be enzymes or receptors found on the surface of the cancer cells. In the case of tumors whose cancer cells express these distinct antigens, these engineered T cells expressing receptors with a variable domain recognizing the tumor-associated antigen, along with a transmembrane-anchoring domain and a T cell receptor activation domain (CAR T cells), are created using cells obtained from the patient’s peripheral blood [189,190]. In this context, the cytolytic actions of CAR T cells are independent of the need for antigen presentation on the human leukocyte antigen molecules [191]. The first generation of CAR T cells did not yield promising results due to poor expansion and low persistence in vivo [192,193]. The next generation CAR T cells now include costimulatory domains (e.g., CD27 and CD28) to enhance the cytocidal and persistence of the engineered T cells [193]. In addition, the fourth generation of CAR T cells now include an IL-12-inducible NFAT expression cassette. Once the CAR T cells recognize the tumor-associated antigen expressed on the cancer cells, the increased production of the proinflammatory cytokine IL-12 results in the activation of downstream signaling and full activation of the CAR T cells to enhance their antitumor functions [194,195].

In BC cells, the development of CAR T cell therapies has been slow. This is mainly due to the lack of breast tumor-associated antigens. In HER2+ BC tumors, for example, the use of anti-HER2− CAR T cell therapy is being considered. Preclinical results from animal experiments look promising in that the use of HER2−CAR T cells decreased primary tumor growth and caused the regression of brain tumor metastasis [196,197]. However, it should be noted that normal breast and other epithelial cells also express the HER2 receptor, although at lower levels. Another example of a tumor-associated antigen in BC is mesothelin, and this is being considered for the generation of the patient’s CAR T cells that would recognize and eliminate the TNBC cells [191,198]. CAR T cell therapies are also being considered in clinical trials. In addition to HER2 and mesothelin, other tumor-associated antigens that are being considered for Phase I/II clinical trials for CAR T cell therapy are MUC1 and MET [199]. Unfortunately, no CAR T cell therapies are yet approved to treat BC tumors.

(C)Cancer vaccines: Vaccines for use as prophylactic measures to prevent tumor development have been developed against viral infections that cause malignancies such the human papilloma virus and the hepatitis B virus [174]. The role of other viruses such the human cytomegalovirus (HCMV) in the development of many malignancies including BC is an active area of research. Recent data indicate that evidence of an HCMV infection can be found in up to 90% of BC patients with expression of the HCMV viral proteins by BC cells [200]. On the other hand, the therapeutic cancer vaccines are still at various stages of development. For example, some prostate cancer cells exhibit overexpression of prostatic acid phosphatase which has led to the development of a vaccine to help the immune system detect and eliminate such prostate cancer cells. Another approach that is being actively considered is the creation of oncolytic viruses where a virus is used to cause forced expression of a toxic protein in cancer cells [174].

BC tumors, in general, have a low mutational burden, making the identification of cancer cell-specific antigens very difficult [198]. However, the immune system does detect abnormally overexpressed proteins such as the HER2 receptor, IGFBP-2, and IGF-IR [201,202]. Among these antigenic proteins, HER2 has been the subject of intense study towards the development of a therapeutic vaccine against HER2+ BCs [202]. Although promising results are observed in clinical trials, these immunotherapies only are effective in 20% of patients, and among the initially responding tumors, sustained responses are a clinical challenge that needs to be addressed. Additional research is needed to provide a framework to develop efficacious immunotherapies for those patients who do not benefit from the currently available immunotherapies.

## 6. Concluding Remarks

Metastasis remains a crucial challenge in treating BC. Understanding the tumor microenvironment of metastatic breast tumors has revealed further heterogeneity in both BC cells and in their microenvironments, including immune responses in metastatic tumors. Unfortunately, immunotherapy has been disappointing. Cancer immunotherapies still hold great promise to offer treatment options for metastatic breast tumors that are currently incurable. To be effective against BC tumors, however, immune suppression needs to be eliminated and better CAR T cells developed to specifically target metastatic breast tumors. Until such technologies are available, biomarkers are needed to identify patients that would benefit from immune checkpoint inhibitors and adoptive cell transfer treatments. Presently, only the expression of PD-L1 in TNBC is used as an indicator for use of checkpoint inhibitors. As we gain insight into the interplay between breast tumors and their metastatic environments, new and effective targeted treatments and immunotherapies will be developed.

## Figures and Tables

**Figure 1 cancers-16-00911-f001:**
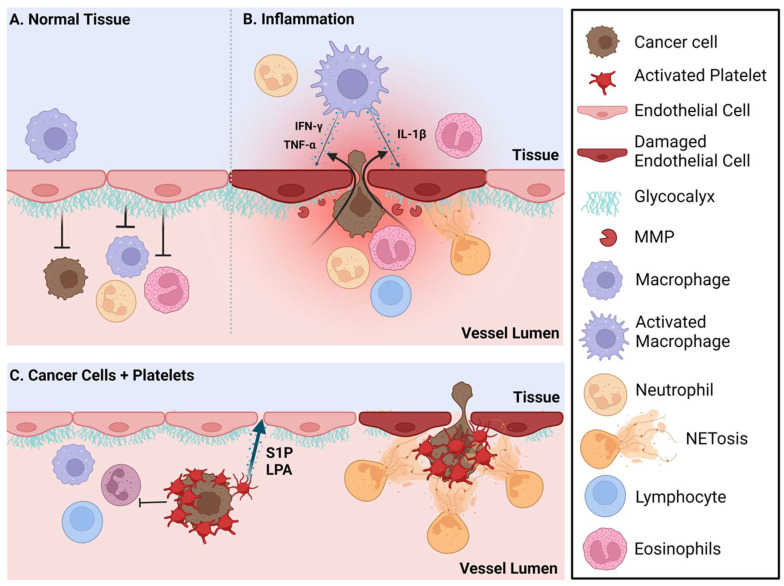
Inflammation and platelet-induced vascular endothelial permeability increases the extravasation of cancer cells into pre-metastatic sites. (**A**) The vessel lumen-facing glycocalyx acts as a barrier that limits extravasation of cancer cells and immune cells. (**B**) During inflammation, the glycocalyx is degraded by matrix metalloproteases (MMPs) and neutrophil extracellular traps (NETs) which increase vascular permeability and cellular extravasation from the vessel lumen. Additionally, during inflammation, activated macrophages in the tissue produce the cytokines TNFα, IFN-γ and IL-1β among others which can further increase endothelial permeability. (**C**) Circulating cancer cells aggregate with platelets, thus avoiding shear stress and immune cell recognition. These platelets are attracted to damaged endothelial cells and will also release sphingosine 1-phosphate (S1P) and lysophosphatidate (LPA), two bioactive lipids that when locally elevated will increase vascular endothelial permeability and cancer cell extravasation. NETs produced by activated neutrophils can also capture the circulating cancer cell/platelet complexes, facilitating their entry into the tissue. Created with BioRender.com.

**Table 1 cancers-16-00911-t001:** Investigational treatments for metastatic breast cancer.

Type of Metastatic Breast Cancer	Intervention/Treatment	Mechanisms of Action	Phase	Identifier
HER2−	Utidelone vs. docetaxel	Microtubule stabilizers	3	NCT05430399
HER2−	Alpelisib in combination with chemotherapy (nab-paclitaxel) and L-NMMA Combination regimen	PI3K inhibitor (alpelisib); microtubule stabilizer (nab-paclitaxel); iNOS inhibitor (L-NMMA);	2	NCT05660083
ER+ HER2−	Enobosarm in combination with abemaciclib	Selective androgen receptor modulator (enobosarm); CDK4/6 inhibitor (abemaciclib)	3	NCT05065411
ER+ HER2−	Combination therapy with anastrozole, fulvestrant, and abemaciclib	Aromatase inhibitor (anastrazole); selective estrogen receptor down-regulator (fulvestrant); CDK4/6 inhibitor (abemaciclib)	2	NCT05524584
ER+ HER2−	ARV-471 in combination with everolimus	Selective estrogen receptor down-regulator (ARV-471); mTOR inhibitor (everolimus)	1	NCT05501769
ER+ HER2−	Gedatolisib plus fulvestrant with or without palbociclib	A dual inhibitor, targets both PI3K and mTOR (gedatolisib), selective estrogen receptor down-regulator (fulvestrant), CDK4/6 inhibitor (palbociclib)	3	NCT05501886
HER2+	YH32367	HER2/4-1BB bispecific antibody (BsAb)	1/2	NCT05523947
HER2+	Tucatinib in combination with pegylated liposomal doxorubicin (Doxil)	HER2 tyrosine kinase inhibitor (tucatinib); DNA intercalation and inhibition of topoisomerase II-driven DNA repair (doxil)	2	NCT05748834
PIK3CA-Mutant HER2+	Combination of alpelisb with tucatinib	PI3K inhibitor (alpelisb); HER2 tyrosine kinase inhibitor (tucatinib);	1/2	NCT05230810
TNBC or HER2+ with brain metastasis	Dendritic cell vaccines against Her2/Her3 and pembrolizumab	Booster of immune response against tumor cells (dendritic cell vaccine); PD-1 receptor monoclonal antibody (pembrolizumab)	2	NCT04348747
TNBC	CDX-301 and CDX-1140 in combination with the standard chemotherapy (pegylated liposomal doxorubicin (Doxil))	Recombinant FMS-like tyrosine kinase 3 ligand (CDX-301); monoclonal antibody as the agonist of CD40 (CDX-1140); DNA intercalation and inhibition of topoisomerase II-driven DNA repair (doxil)	1	NCT05029999
TNBC	ASTX727 (cedazuridine, decitabine) to chemotherapy (paclitaxel) and immunotherapy (pembrolizumab)	ASTX727 composed of decitabine as a hypomethylating agent protected against deamination by the cytidine deaminase inhibitor component, cedazuridine; microtubule stabilizer (paclitaxel); PD-1 inhibitor (pembrolizumab)	1	NCT05673200
TNBC refractory to anthracycline with PI3KCA or PTEN alterations	Alpelisib in combination with nab-paclitaxel	PI3K inhibitor (alpelisib); microtubule stabilizer (nab-paclitaxel);	2	NCT04216472
TNBC with either PI3KCA mutation or PTEN loss	Alpelisib in combination with nab-paclitaxel	PI3K inhibitor (alpelisib); microtubule stabilizer (nab-paclitaxel);	3	NCT04251533
MUC1* positive breast cancer	Autologous huMNC2-CAR44 T cells	Chimeric antigen receptor (CAR)-modified T cells that target specifically the cancerous form of cleaved MUC1 (called MUC1*), which is known as a growth factor receptor of many solid tumors.	1	NCT04020575

**Table 2 cancers-16-00911-t002:** Immunotherapy clinical trials in metastatic breast cancer. • = designate if multitherapy or mono-therapy was delivered.

	Breast Cancer Stage	Immunotherapeutic			Therapy Type		Reference
		PD-L1 Inhibitor	PD-1 Inhibitor	CTLA-4 Inhibitor	Mono-Therapy	Multi-Therapy	
Phase 1	Early						
		Atezolizumab				•	NCT03802604
	Locally Advanced						
		Atezolizumab				•	NCT03800836
		Durvalumab				•	NCT03356860
		M7824			•		NCT02699515
			Pembrolizumab			•	NCT03310957
	Metastatic						
		Atezolizumab				•	NCT03853707
		Avelumab				•	NCT04360941
			Nivolumab			•	NCT02393794
			Pembrolizumab			•	NCT03362060 NCT03272334
	Not Specified						
			Pembrolizumab			•	NCT06246968
Phase 2	Early						
		Avelumab				•	NCT04841148
			Pembrolizumab			•	NCT05675579
	Locally Advanced						
		Atezolizumab				•	NCT02924883 NCT03424005
			Pembrolizumab			•	3
	Metastatic						
		Atezolizumab				•	NCTT0294883
		Avelumab				•	NCT04215146 NT03147287
				Ipilimumab		•	NCT03789110
			Nivolumab			•	NCT03316586
			Pembrolizumab		•	•	NCT03139851 NCT02447003
	Not Specified						
		Atezolizumab				•	NCT03170960
				Ipilimumab		•	NCT03815890
			Nivolumab			•	NCT03815890 NCT03742968
			Pembrolizumab			•	NCT03025035
Phase 3	Early						
		Atezolizumab				•	NCT03726879 NCT03595592
			Nivolumab			•	NCT04109066
			Pembrolizumab			•	NCT03725059
	Locally Advanced						
		Atezolizumab				•	NCT04148911 NCT03125902
			Pembrolizumab			•	NCT05382286 NCT03036488
	Metastatic						
		Atezolizumab				•	NCT04177108 NCT04740918

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
