# Peer review of "Breast Tumor Metastasis and Its Microenvironment: It Takes Both Seed and Soil to Grow a Tumor and Target It for Treatment"

_cancers, 2024, doi:10.3390/cancers16050911_

Round 1
Reviewer 1 Report
Comments and Suggestions for Authors
This is an interesting article aimed at providing the reader with a review of breast cancer heterogeneity and standard of care plus a discussion of innovative experimental treatments, starting from the seed and soil concept. The article is well organized, however, this reviewer believes it can be improved and made more interesting
Major comments
paragraph 5.1 this section is lengthy and should focus more on investigational treatments and their rationale and results than on standard of care
paragraph 5.2 same as above, in this reviewer’s opinion there should be more description and discussion of investigational treatments than history and basics of immunotherapy from line 600 to 642
Minor comments
Figure 1 legend is split between lines 223 on page 5 and up to line 229 on page 6, and lines 293 to 305 on page 7. Please reformat
Table 1 the title would better read “Investigational treatments …”
Table 2 this reviewer’s suggestion would be to include for each trial a reference to the accession number on clinicaltrials.gov similar to table 1. Also, the format of the 2 tables should be more or less the same
page 2 line 52 no keywords are provided
page 6 line 247 there’s a space between “growth.” and “A recent”
page 6 line 261 please insert a space between “stress” and “[75]”
page 7 line 316 please insert a space between “BC” and “[88,89]”
page 10 line 396 please reformat
page 11 line 419 “accumulate” is likely “accumulation”
page 13 line 515 the accession number for phase 3 clinical trial with enobosarm is different than the one listed in Table 1
page 13 line 528-529 “This issue has been addressed” should be “is being addressed due to the fact that the clinical efficacy of DC vaccines is not yet demonstrated. Also, “through recruitment of DC vaccines” should be rephrased
page 16 lines 642-644 also atezolizumab (anti-PD-L1) is approved for treatment of TNBC
page 18 line 733 “cacner” should be “cancer”
Author Response
We would like to thank the reviewers’ comments. The comments improved the review manuscript and a detail response to the specific comments are listed below.
Reviewer #1
Major comments
paragraph 5.1 this section is lengthy and should focus more on investigational treatments and their rationale and results than on standard of care
Response: We have revised section 5.1 to focus more on investigational treatments.
paragraph 5.2 same as above, in this reviewer’s opinion there should be more description and discussion of investigational treatments than history and basics of immunotherapy from line 600 to 642
Response: We have revised section 5.2 to focus more on investigational treatments.
Minor comments
Figure 1 legend is split between lines 223 on page 5 and up to line 229 on page 6, and lines 293 to 305 on page 7. Please reformat
Response: We have reformatted Figure. 1
Table 1 the title would better read “Investigational treatments …”
Response: We have changed the title for Table 1 as requested.
Table 2 this reviewer’s suggestion would be to include for each trial a reference to the accession number on clinicaltrials.gov similar to table 1. Also, the format of the 2 tables should be more or less the same
Response: Most trails are ongoing and do not have a reference.
page 2 line 52 no keywords are provided
Response: Keywords were added.
page 6 line 247 there’s a space between “growth.” and “A recent”
Response: The text was edited.
page 6 line 261 please insert a space between “stress” and “[75]”
Response: The text was edited.
page 7 line 316 please insert a space between “BC” and “[88,89]”
Response: The text we edited.
page 10 line 396 please reformat
Response: It was reformatted.
page 11 line 419 “accumulate” is likely “accumulation”
Response: Text was edited.
page 13 line 515 the accession number for phase 3 clinical trial with enobosarm is different than the one listed in Table 1
Response: Text was revised.
page 13 line 528-529 “This issue has been addressed” should be “is being addressed due to the fact that the clinical efficacy of DC vaccines is not yet demonstrated. Also, “through recruitment of DC vaccines” should be rephrased
Response: Text was edited.
page 16 lines 642-644 also atezolizumab (anti-PD-L1) is approved for treatment of TNBC
Response: Text was edited
page 18 line 733 “cacner” should be “cancer”
Response: Text was edited

Reviewer 2 Report
Comments and Suggestions for Authors
The article entitled “Breast tumor metastasis and its microenvironment: It takes 2 both seeds and soil to grow a tumor and target it for treatment”, submitted by Dr. Bonni and colleagues, represents an excellent and very well updated review on the topic. Although it is possible to find several recent reviews on BC and its microenvironment, this one addresses this complex subject in a detailed and very understandable way. The authors explain the complexity of the heterogeneity of both tumor cells and their microenvironment, factors that strongly contribute to the emergence of metastases and resistance to treatments. I just missed any comment on the contribution of extracellular vesicles in defining the pre-metastatic site. But, overall, the article is scientifically impeccable.
There are some formatting or mistyping errors:
1. Lines 28-29: Breast tumor grows, or breast tumors grow.
2. Line 57: Wording: cancer-causing cancer-related mortality in women.
3. There are no keywords.
4. The entire article is based on the concept of seed and soil but the authors do not make any reference to the author of this theory, Steven Paget.
5. Line 143, please delete the comma after “By the time…”
6. There is no citation for figures 1 and 2 in the text, nor for the tables, as there should be. The citation for the figures appears in the section subtitle but is not the appropriate place.
7. Caption to Figure 1 is repeated in line 223 and misplaced.
8. Line 176. A number “3” is missing.
9. Lines 396-397 are cut.
10. Line 733. It should be “…trials…..cancer
Comments on the Quality of English LanguageNo further comments
Author Response
We would like to thank the reviewers’ comments. The comments improved the review manuscript and a detail response to the specific comments are listed below.
Reviewer #2
I just missed any comment on the contribution of extracellular vesicles in defining the pre-metastatic site. But, overall, the article is scientifically impeccable.
Response: We have added a paragraph in the review discussing extracellular vesicles and breast cancer metastasis.
There are some formatting or mistyping errors:
- Lines 28-29: Breast tumor grows, or breast tumors grow.
Response: Text was edited
- Line 57: Wording: cancer-causing cancer-related mortality in women.
Response: Text was edited
- There are no keywords.
Response: Keywords were added.
- The entire article is based on the concept of seed and soil but the authors do not make any reference to the author of this theory, Steven Paget.
Response: Reference was added.
- Line 143, please delete the comma after “By the time…”
Response: Text was edited
- There is no citation for figures 1 and 2 in the text, nor for the tables, as there should be. The citation for the figures appears in the section subtitle but is not the appropriate place.
Response: Figure citations in text were added.
- Caption to Figure 1 is repeated in line 223 and misplaced.
Response: Figure 1 was reformatted.
- Line 176. A number “3” is missing.
Response: Text was edited.
- Lines 396-397 are cut.
Response: Text was edited.
- Line 733. It should be “…trials…..cancer
Response: Text was edited.

Round 2
Reviewer 1 Report
Comments and Suggestions for Authors
The authors have improved the the manuscript
however, please note that
page 13 line 401 following "endothelium" there is an empty space and the sentence continues on the next page
figure 3 panel A there is a red bar crossing the entire figure, I do not understand what it is
on page 25 Table 2 has neither reference to a NCT number from clinicaltrials.gov nor to a publication. In the interest of readers, please provide either one or, ideally, both
on page 17 lines 539-541 the sentence about sacituzumab govitecan under consideration should be rephrased as this antibody drug conjugate has received marketing authorization from both EMA and FDA in breast cancer
page 17 lines 528-529 "has being" should be "is being"
Author Response
Thank you for the comments. Here are the minor corrections.
page 13 line 401 following "endothelium" there is an empty space and the sentence continues on the next page
Response: This has been corrected.
figure 3 panel A there is a red bar crossing the entire figure, I do not understand what it is
Response: This has been corrected.
on page 25 Table 2 has neither reference to a NCT number from clinicaltrials.gov nor to a publication. In the interest of readers, please provide either one or, ideally, both
Response: NCT numbers have been added to the table.
on page 17 lines 539-541 the sentence about sacituzumab govitecan under consideration should be rephrased as this antibody drug conjugate has received marketing authorization from both EMA and FDA in breast cancer
Response: This sentences has bee rephrased.
page 17 lines 528-529 "has being" should be "is being"
Response: This has been corrected.